# Rodent Models of Post-Stroke Dementia

**DOI:** 10.3390/ijms231810750

**Published:** 2022-09-15

**Authors:** Hahn Young Kim, Dong Bin Back, Bo-Ryoung Choi, Dong-Hee Choi, Kyoung Ja Kwon

**Affiliations:** 1Department of Neurology, Konkuk University Medical Center, Konkuk University School of Medicine, 120-1 Neungdong-ro, Gwangjin-gu, Seoul 05030, Korea; 2Department of Medicine, Konkuk University School of Medicine, Seoul 05030, Korea

**Keywords:** post-stroke dementia, rodent model, ischemic stroke, chronic cerebral hypoperfusion, amyloid deposits, intracerebral hemorrhage, subarachnoid hemorrhage

## Abstract

Post-stroke cognitive impairment is one of the most common complications in stroke survivors. Concomitant vascular risk factors, including aging, diabetes mellitus, hypertension, dyslipidemia, or underlying pathologic conditions, such as chronic cerebral hypoperfusion, white matter hyperintensities, or Alzheimer’s disease pathology, can predispose patients to develop post-stroke dementia (PSD). Given the various clinical conditions associated with PSD, a single animal model for PSD is not possible. Animal models of PSD that consider these diverse clinical situations have not been well-studied. In this literature review, diverse rodent models that simulate the various clinical conditions of PSD have been evaluated. Heterogeneous rodent models of PSD are classified into the following categories: surgical technique, special structure, and comorbid condition. The characteristics of individual models and their clinical significance are discussed in detail. Diverse rodent models mimicking the specific pathomechanisms of PSD could provide effective animal platforms for future studies investigating the characteristics and pathophysiology of PSD.

## 1. Introduction

Stroke is one of the most common causes of death together with cancer and cardiovascular disease in most developed countries [1]. Residual physical disability and cognitive decline are the most common complications among stroke survivors [2]. Even in physically independent stroke survivors, cognitive decline can be a major hurdle when they try to return to normal everyday life or premorbid social life. The prevalence of post-stroke cognitive dysfunction, ranging from mild cognitive impairment to dementia, can be as high as 80% [3]. By definition, “post-stroke” dementia (PSD) implies that dementia occurred after stroke. Although the concept of PSD seems simple and easy to define, the clinical diagnosis and classification of PSD are not always simple and easy because of its etiological heterogeneity. The clinical manifestations of PSD can vary from simple manifestations of underlying pre-stroke cognitive decline to newly developed dementia resulting from current stroke or any complicated interaction with diverse comorbid conditions. In a systematic review and meta-analysis, the prevalence of pre-stroke dementia ranges from 9.1% in population-based studies to 14.4% in hospital-based studies, whereas the prevalence of PSD ranges more variably from 7.4% to 41.3% [4].

PSD may be related to diverse vascular dementia-associated conditions, such as strategic infarct dementia, multi-infarct dementia, subcortical ischemic vascular dementia, hypoperfusion dementia, mixed dementia, or vascular dementia with underlying Alzheimer’s disease (AD) pathology. For example, in strategic infarct dementia, the left frontotemporal lobes, left thalamus, and right parietal lobe were strongly associated with the development of PSD [5]. In a systemic review and meta-analysis study, some blood proteins have been suggested as potential biomarkers for PSD [6]. One-fourth of stroke patients develop PSD within three months of the stroke episode [7]. In some patients with PSD, underlying dementia may be aggravated after stroke or slowly deteriorate as a delayed-onset type [2]. Clinical manifestations and underlying pathomechanisms may be heterogeneous in PSD depending on the subtype [8]. Strategic infarct dementia or multi-infarct dementia that develop after stroke can be considered early-onset PSD, whereas delayed-onset PSD can slowly deteriorate after stroke in patients with underlying premorbid conditions [8].

Stroke survivors in the chronic phase who are more than five months post-stroke and a mouse model at six months post-photothrombotic stroke showed similar chronic cognitive impairments [9]. New dementia develops after stroke in some patients, whereas pre-symptomatic dementia may progress to overt dementia in others. Various risk factors or undermining conditions predisposed to PSD have been suggested in the literature [10]. However, experimental research using animal models of PSD encompassing these diverse clinical situations has not been well discussed.

The premorbid conditions for PSD can vary. Concomitant vascular risk factors, such as aging, diabetes mellitus, hypertension, and dyslipidemia, can predispose patients to PSD. Underlying pathological brain conditions, such as chronic cerebral hypoperfusion or AD pathology, can lead to the development of PSD. Subcortical lesions with white matter hyperintensities are associated with chronic cerebral hypoperfusion dementia [11]. AD pathology, such as amyloid deposits, can be observed in ischemic lesions or the penumbra of stroke patients [12].

A rodent model for PSD can be a specific type of vascular dementia model resulting from any compromised conditions induced by stroke in its broad term [10]. Although some studies using bilateral common carotid artery occlusion (BCCAo) models with various dyeing methods have shown pathological lesions that are responsible for cognitive impairment [13], chronic vascular insufficiency models without explicit acute infarcts such as BCCAo, unilateral carotid artery occlusion, or stenosis models with various methods were excluded from this review because of the absence of explicit acute ischemic injury in these models. In addition, rodent models, which have shown significant cognitive impairment in behavioral tests, were considered appropriate PSD models. First, we selected studies that showed evident acute stroke lesions with diverse experimental methods, including filament suture, photothrombotic, and stereotactic injection models. Second, cognitive impairment should be confirmed by various behavioral tests in post-stroke conditions. We reviewed diverse rodent models that mimic PSD with or without concomitant premorbid conditions. Conceptualized figures of the diverse rodent models of PSD are displayed in Figure 1, and its comparative clinical PSD subtypes were also postulated in the associated clinical conditions (Table 1). In addition, the pathophysiology of PSD was investigated, and comparable clinical situations of diverse rodent models of PSD were discussed. Future directions of experimental studies using rodent models of PSD have been suggested.

## 2. Rodent Models

We searched eligible rodent models for PSD on the basis of the pre-described criteria. Owing to the diversity and heterogeneity of the selected models, we classified these models into three categories, namely, surgical technique, special structure, or comorbid condition. The “surgical technique” category focuses on the modeling technique such as middle cerebral artery occlusion (MCAO), the combination of BCCAo and MCAO, and subarachnoid hemorrhage (SAH) models. The “special structure” category focuses on the structures related to cognition such as cortical stroke, hippocampal stroke, lacunar stroke, and intracerebral hemorrhage (ICH) models. The “comorbid condition” category includes stroke models with comorbid conditions or with underlying AD pathology.

### 2.1. MCAO Model

Cognitive impairments have been reported in many experiments using MCAO models [14,15,16,17,18,19,20,21]. However, most experiments using animal models of acute infarcts have focused on the pathophysiology of the acute stroke phase. Behavioral tests mainly focus on the recovery of sensorimotor deficits after stroke. If cognitive behavioral tests were performed in the acute stroke phase, the results of the tests might have been biased by the underlying sensorimotor deficits.

Unlike the human cognitive test, minor sensorimotor impairments may undermine cognitive function. The Morris water maze task or novel object test also requires a certain level of sensorimotor function to execute tasks successfully. Therefore, some results of the behavioral cognitive test, which are usually performed less than four weeks after stroke, might have been contaminated by minor sensorimotor dysfunction [14,15,16,17,18,19,20,21].

In principle, cognitive impairments should be investigated after full recovery from post-stroke sensorimotor deficits. Although the water maze task was performed less than four weeks after stroke onset, normal swimming speed might suggest the full recovery of post-stroke neurological deficits, and the results of the cognitive consequences might be acceptable. Some experiments have shown significant cognitive impairments in the chronic phase of stroke even after the full recovery of the sensorimotor neurological deficits [22,23]. A PSD model using MCAO might mimic PSD after a large hemispheric infarction with concomitant neurological deficits. In case cognitive impairment is due to the involvement of a large brain area, large hemispheric infarct dementia and multiple territorial infarct dementia may share some clinical similarities. The associated clinical conditions with a PSD model using MCAO may be intra- or extracranial atherosclerosis or embolic stroke originating from diverse sources.

### 2.2. Stroke Model with Comorbid Conditions

Although the MCAO model might have shown some cognitive impairments in common with PSD after large territorial infarcts, most experiments have been performed using young and healthy animals [14,15,16,17,18,19,20,21]. However, in the clinical setting, the development of PSD prefers underlying comorbid conditions such as aging, hypertension, diabetes, or dyslipidemia [11]. Aged rats with MCAO are more vulnerable to cognitive impairment after MCAO than younger rats [23]. More cognitive impairments associated with grey matter atrophy and apoptosis, neuroinflammation, and DNA damage were observed in hypertensive MCAO rats than in normotensive MCAO rats [28]. In other predisposing conditions such as obesity, diabetes mellitus, and dyslipidemia, more cognitive impairments have been observed when MCAO is performed [17,18,21,29,30]. In clinical aspects, these MCAO models in diverse predisposing conditions might be more plausible for PSD induced by territorial infarcts that usually occur in diverse premorbid conditions. These diverse premorbid conditions can work synergistically to enhance PSD in the clinical situation. Proinflammatory microglial activation and cognitive aggravation were reported not only in an MCAO model but also in an endothelin-1 (ET-1) injection model with diabetes mimicking conditions via the infusion of a high glucose solution [51]. Therefore, diverse comorbid conditions can work universally as a synergistic factor in diverse rodent models of PSD.

### 2.3. Cortical Stroke Model

Cognitive impairments in a cortical stroke model, which has an infarct size that is usually smaller than that of an MCAO model, have been investigated. A photothrombotic model of cortical arterial branch occlusion that was used to investigate the pathological consequences of focal ischemia has also shown some cognitive impairments [38,39,40,41]. A smaller infarct size that mainly focuses on the sensorimotor cortex might be the reason why cognitive impairments were minimal. Owing to the convenience of vascular targeting, the photothrombotic stroke model is useful for investigating the cognitive impairments induced by topographic brain lesions in the bilateral frontal cortices [31,32], medial prefrontal cortex [24,35,42], and right medial agranular cortex, which is responsible for unilateral spatial neglect [36]. Secondary neurodegeneration in the hippocampus has been reported to be responsible for post-stroke cognitive impairment following photothrombotic stroke in motor and somatosensory cortical strokes [33]. Chronic stress is also an aggravating factor for secondary thalamic amyloid accumulation as secondary neurodegeneration following photothrombotic stroke [43]. Interestingly, multi-infarct dementia in humans was simulated by an experiment showing that recurrent photothrombotic cortical infarcts in mice develop progressive cognitive impairments when recurrent cortical infarcts are induced [25].

The focal parenchymal injection of ET-1 can simulate small cortical infarcts in humans [34,37,44]. Small cortical infarcts induced by ET-1 injection in the frontal cortex are asymptomatic in motor and cognitive behavioral tests [34]. Unilateral or bilateral injections of ET-1 in the prefrontal cortex showed some cognitive impairments in mice [44] and rats [37]. To mimic anterior cerebral artery infarction in humans, ET-1 injection into the anterior cerebral artery in rats induced cognitive and executive dysfunctions [24,26]. Cognitive impairments in these rodent models with lesioning frontal cortex might suggest frontal lobe dysfunction, including working memory in humans, due to strategic infarct dementia. Asymptomatic infarct lesions are frequently observed in the clinical setting. Small cortical lesions may not lead to cognitive impairment or cause subclinical cognitive impairment if they occur in the non-eloquent brain area. When stroke happens in the eloquent cortex or key cortical area associated with cognition, it could have devastating effects even if it is a very small cortical infarct.

### 2.4. Hippocampal Stroke Model

The blood supply to the human hippocampus is mostly delivered by vertebrobasilar circulation. In rodents, the blood supply to the hippocampus is largely supplied by deep cerebral arteries arising from the internal carotid artery [47]. MCAO or global ischemia models, occluding both internal carotid arteries, can also show hippocampal damage. Hippocampus-dependent spatial memory can deteriorate in these models [14,15,16,18,19,20,27]. Diverse hippocampal pathologies associated with cognitive impairment in the MCAO model have been reported, including hippocampal inflammation [14,19,20], the suppression of long-term potentiation [15], impaired neurogenesis [16], neuronal degeneration [18], and apoptosis [27].

Focal hippocampal infarcts induced by ET-1 injection [48] or photocoagulation [45] have shown some cognitive impairment. Bilateral injection of ET-1 into the hippocampus showed cognitive impairment only when combined with a predisposing state of stress [46]. Given that the hippocampus is an essential part of cognition, these rodent models may share some characteristics of strategic infarct dementia in humans. In addition to the involvement of the hippocampus, which is the main center of spatial memory in rodents, the involvement of other brain areas participating in the Papez circuit such as the thalamus, mammillary body, subiculum, and cingulate gyrus can result in the development of PSD even after a very small regional stroke.

### 2.5. Lacunar Stroke Model

Stereotactic parenchymal injection of ET-1 into the striatum [52,53,54,55,56], thalamus [49], and periventricular white matter [57] has been studied to simulate lacunar infarcts in humans. Most studies using a single lacunar infarct model have not performed behavioral tests [49,53,55] or performed sensorimotor tests only [52,54,56]. A single lacunar infarct in a rodent model might not be sufficient to induce PSD on the basis of its location or lesion volume.

Even a single lacunar infarct can cause significant cognitive impairment when it involves a very critical location in complicated human cognitive pathways such as the thalamus, corpus callosum, or hippocampus. In rodents, a single small infarct may not be critical enough to induce cognitive impairment because of their simpler cognitive circuit. Multiple ET-1 injections have been used to mimic cognitive impairment with multiple lacunes [50]. ET-1 injection into the bilateral thalamus did not exhibit cognitive impairment, whereas injection into the bilateral medial prefrontal cortex showed some cognitive impairments [50]. Although single or even multiple lacunar infarcts by ET-1 injection might not have induced explicit cognitive impairments, these models can suggest predisposing conditions, such as white matter hyperintensities or multiple lacunar infarcts with underlying small vessel disease, that are vulnerable to the development of PSD. The multiple ET-1 injection model has been considered a pre-clinical model of human cerebral small vessel disease [58]. A single ET-1 injection model mimics a single lacunar infarct. Its main pathophysiology is the arteriolosclerosis of a single perforator artery associated with hypertensive angiopathy. A multiple ET-1 injection model can mimic multiple lacunar infarct states, suggesting a more extensive arteriolosclerosis of multiple perforating arteries. Multiple lacunar status is one of the common types of small vessel disease conditions and is a predisposing condition for vascular dementia, especially subcortical ischemic vascular dementia.

### 2.6. Combining BCCAo and MCAO Model

Rodent models with surgical procedures intervening in cerebral perfusion, such as permanent occlusion or stenosis of the bilateral carotid arteries, can simulate vascular dementia induced by chronic cerebral hypoperfusion [64,65]. However, these models cannot be classified as PSD because definite acute infarcts were not induced. Therefore, animal models of chronic cerebral hypoperfusion will not be discussed further in the current study. Instead, we suggest a rodent model that combines BCCAo and MCAO as a unique rodent model for PSD [66,67].

Two sequential surgical interventions separated by a two-week interval were performed: (1) transient MCAO surgery to mimic a territorial infarct, and (2) BCCAo surgery to mimic chronic cerebral hypoperfusion. Cognitive behavioral tests, including the Morris water maze task and the novel object test, were performed when swimming speed and modified neurological severity scores were nearly normalized eight weeks after surgical intervention [66,67]. Therefore, impairments during these tests could be confirmed as a result of cognitive dysfunction and not neurological motor dysfunction. Among four groups (two by two surgical interventions), spatial memory measured by search error, time latency, path length, and probe trial were synergistically impaired when BCCAo was superimposed on MCAO [66]. Enhanced neuroinflammation with astroglial or microglial activation and amyloid pathology were observed in the ipsilateral cortex, thalamus, and hippocampus when BCCAo was superimposed on MCAO [66]. Rats with BCCAo have been considered a rodent model for vascular dementia induced by chronic cerebral hypoperfusion [78,79]. Considering that chronic cerebral hypoperfusion induced by long-standing hypertension or severe carotid stenosis can lead to ischemic white matter injury [80], cognitive impairments in BCCAo rats evidenced by white matter disintegration [78,79] may be due to chronic cerebral hypoperfusion, which is one of the well-known predisposing conditions for the development of PSD. In addition to white matter disintegration in BCCAo rats, the glymphatic pathway, which has been noticed as an interstitial space fluid bulk flow clearance for brain metabolic wastes such as amyloid, tau, and synuclein [81], could be also disturbed by the bilateral ligation of common carotid arteries. Cerebral arterial pulsation mainly transmitted through the carotid arteries has been revealed as a driving force to maintain glymphatic clearance [81]. Our experiments showed that the permanent ligation of both common carotid arteries in a BCCAo model may have attenuated arterial pulsation in cerebral arteries, thus resulting in the perturbation of the glymphatic pathway [66,67]. Glymphatic pathway-related aquaporin4 (AQP4) distribution changed from perivascular to parenchymal pattern [66], thus suggesting that a perturbed glymphatic pathway due to the aberrant dislocation of the AQP4 water channel along with neuroinflammation may have contributed to the development of amyloid deposits in the post-stroke state.

Enhanced amyloid deposits in combination with the BCCAo and MCAO models may be due to glymphatic dysfunction induced by chronic cerebral hypoperfusion with BCCAo surgery [66,67]. A rodent model combining BCCAo and MCAO that can mimic a clinical setting in which chronic cerebral hypoperfusion or glymphatic dysfunction is superimposed with stroke has shown that compromised predisposing conditions might be critical to the development of PSD [66,67]. This unique rodent model combining BCCAo and MCAO could work as a rodent experimental platform for the evaluation of PSD associated with white matter hyperintensities, leukoaraiosis, or impaired glymphatic clearance.

### 2.7. Stroke Model with Underlying AD Pathology

A combined rat model of striatal injection of ET-1 and amyloid beta toxicity was developed to investigate the comorbid conditions of acute ischemic injury and amyloid pathology [59,60,61,62,63]. The infarct volume increases, and cognitive deficits deteriorate progressively in the presence of high levels of amyloid [61,62,63]. ET-1 induced stroke in transgenic amyloid precursor protein mice, thus simulating the interaction between stroke and amyloid pathology and showing that subclinical stroke could induce an increase in localized amyloid pathology [68]; this is consistent with the clinical finding that stroke patients with underlying amyloid pathology could develop a more severe and rapid cognitive decline over three years than those without underlying amyloid pathology [82]. AD-related pathology has been reported in ischemic lesions or the surrounding penumbra in animal experiments [83,84,85]. Instead of the simple direct effect of the infarct itself, complex interaction affecting amyloid metabolism, including enzymatic degradation, receptor-mediated blood–brain barrier clearance, cerebrospinal fluid absorption clearance, and interstitial fluid bulk flow clearance [81] can contribute to the development of PSD on the basis of the perturbation of the underlying AD pathology. Underlying conditions, such as white matter hyperintensities, hypertensive angiopathy, and baseline amyloid deposit in the brain parenchyma and vessels, can be a predisposition toward the development of PSD [10]. Therefore, these animal models are compatible in a clinical setting in which PSD develops in stroke patients with underlying subclinical or clinical AD pathology.

### 2.8. ICH Model

Post-hemorrhagic stroke cognitive impairment can also be considered as PSD [10,86]. However, cognitive impairment among hemorrhagic stroke survivors has not been studied well. Moreover, animal experiments focused on post-hemorrhagic stroke dementia are scarce. ICH is one of the most common subtypes of hemorrhagic stroke. In clinical settings, post-ICH cognitive impairments have been observed among acute phase survivals up to 80% (usually less than four weeks), decreased to 40% at three months, and gradually increased again during long-term follow-up [87,88]. In addition to hematoma size and location [89], underlying conditions such as aging [87], small vessel disease [90], education level [91], diabetes [92,93], or the presence of previous stroke or cerebral amyloid angiopathy [94] have been reported as provocation factors.

Compared with the simple parenchymal injection model of collagenase, donor, or autologous blood [69,70,71], transgenic models such as cerebral amyloid angiopathy or spontaneously hypertensive stroke-prone rats may be more compatible with the pathogenesis of ICH [71]. Behavioral tests have focused on sensorimotor recovery, mostly in the acute stage. Some studies performed in the chronic stage (more than five weeks after experimental ICH) showed cognitive impairments after the full recovery of sensorimotor functions [72,73,74]. Impaired long-term potentiation in hippocampal neurons [72], white matter injury with iron deposition, and neuroinflammation [73] have been suggested to be responsible for delayed cognitive impairments after ICH. Hypertensive angiopathy and cerebral amyloid angiopathy can be the main pathophysiology of ICH. Considering frequent underlying small vessel diseases in patients with ICH, clinical conditions, such as multiple lacunar status, white matter hyperintensities, or multiple microbleeds, should be considered when designing a PSD animal model with ICH. The long-term cognitive outcome and pathomechanism of the chronic phase after ICH need to be investigated, especially with respect to PSD.

### 2.9. SAH Model

Although another common subtype of hemorrhagic stroke is SAH, the clinical significance of post-SAH cognitive impairment has rarely been investigated. Diverse SAH animal models, such as endovascular filament perforation, cisterna magna injection, and cortical surface injection, have been most popularly introduced [75,76,77]. Although cognitive behavioral aspects have rarely been evaluated in most experimental studies, the Morris water maze, elevated T maze, and elevated plus maze tests have been used in some studies [76]. In addition to vasospasm, diverse mechanisms of brain injury, such as cortical spreading ischemia, micro thrombosis with microcirculation failure, blood–brain barrier damage, and increased intracranial pressure, have been introduced [76,77]. However, a more specific mechanism focusing on cognitive impairment after SAH needs to be investigated. When designing a PSD animal model with SAH, clinical conditions, such as aneurysmal rupture, arteriovenous malformation, and superficial cortical hemosiderosis, need to be considered.

## 3. Challenges: Investigating the Pathophysiology of PSD

Considering that the pathophysiology of dementia, including vascular dementia, AD, and mixed dementia, is heterogeneous [95], the nature of PSD could be complicated and heterogeneous. Diverse PSD, including strategic infarct dementia, multi-infarct dementia, subcortical ischemic vascular dementia, hypoperfusion dementia, or mixed dementia, could be more or less related to the vascular component rather than pure neurodegenerative components, such as pure AD [96].

Similar to the clinical heterogeneity of PSD, various animal models for PSD have their own usefulness in investigating the diverse pathophysiology of PSD (Table). Post-MCAO, cortical infarction, hippocampal stroke, and multiple injection models carry special characteristics of PSD after large territorial, small cortical, strategic, and multiple lacunar infarcts, respectively. AD-related pathologies, such as amyloid deposits, can be observed in ischemic lesions or the surrounding penumbra in animal experiments [83,84,85,97] or in patients with stroke [12]. Increased amyloid deposition after stroke may be due to not only the increased production of amyloid but also to the decreased clearance of amyloid. The glymphatic clearance of amyloid can be attenuated by underlying chronic cerebral hypoperfusion [66,67] or stroke [98], thus resulting in amyloid deposition. Concomitant chronic cerebral hypoperfusion or amyloid pathology has been evaluated as an underlying devastating condition that provokes PSD. Remote amyloid deposits in the ipsilateral thalamus after MCAO in rats have been reported as experimental evidence supporting secondary neurodegeneration with amyloid pathology after stroke [85,99,100,101]. In our previous study [66], secondary neurodegeneration with amyloid pathology was significantly enhanced when stroke occurred with cerebral hypoperfusion. Amyloid deposits observed in ischemic lesions and the surrounding penumbra [83,84,85,97] or in the ipsilateral thalamus, hypothalamus, or midbrain after MCAO as remote secondary neurodegeneration [85,99,100,101] suggests the ample contribution of the amyloid pathology to the development of PSD. In addition to amyloid-related neurodegeneration, diffuse brain atrophy after stroke could contribute to the development of PSD, as evidenced by the experiment showing progressive secondary diffuse cortical atrophy after ET-1 injection into the sensorimotor cortex [102].

Immunological mechanisms should be considered as a link between inflammation and PSD. Chronic brain inflammation followed by stroke could provoke the inefficient clearance of myelin debris and a prolonged innate and adaptive immune response, thus leading to immunosuppression, which increases the risk of poststroke infection and subsequent immune activation [103]. Therefore, controlling post-stroke immunomodulation should be considered a promising candidate to prevent PSD.

Instead of the direct effects of stroke, more complex interactions with underlying white matter lesions, hypertensive angiopathy, and amyloid deposits in the brain parenchyma and vessels could contribute to the final development of PSD [10]. The underlying conditions associated with vascular risk factors, such as aging, hypertension, and diabetes, have been studied as synergistic factors that induce or aggravate PSD. The unknown pathophysiological interactions between PSD and glymphatic pathway or imaging markers of the small vessel disease, such as enlarged perivascular spaces, multiple microbleeds, or superficial cortical hemosiderosis, are challenging topics to be investigated in the future.

## 4. Future Directions

Owing to the heterogeneous characteristics of PSD, animal models should be customized to the correct type of PSD depending on the focus of the study. The MCAO model and lacunar stroke model, which are the most commonly used rodent models for studying ischemic pathophysiology, are also useful for studying PSD. In addition, underlying conditions, such as comorbid diseases or pathologic conditions, should be considered in specific animal models for PSD. For example, we have proposed an animal model that combines BCCAo and MCAO to emphasize the effect of chronic cerebral hypoperfusion on the development of PSD. The experimental design should also include an appropriate behavioral cognitive test that can be evaluated independently of sensorimotor dysfunction, which is the main consequence of stroke. Diverse animal models mimicking the specific pathomechanisms of PSD could provide effective animal platforms for future studies that investigate the characteristics and pathophysiology of PSD.

## Figures and Tables

**Figure 1 ijms-23-10750-f001:**
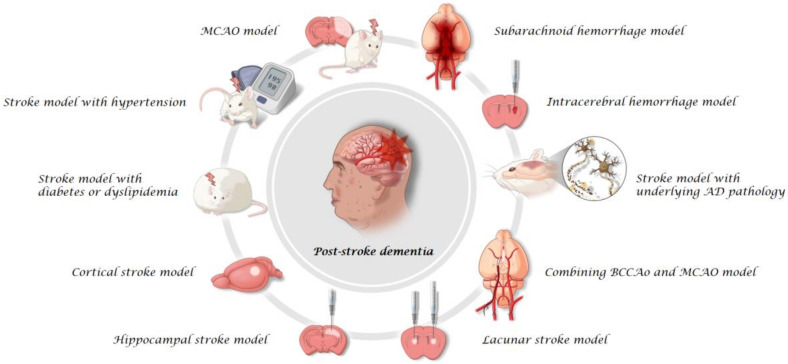
Diverse rodent animal models of post-stroke dementia. (Middle cerebral artery occlusion, MCAO; bilateral common carotid artery occlusion, BCCAo; Alzheimer’s disease, AD).

**Table 1 ijms-23-10750-t001:** Rodent animal models of post-stroke dementia (PSD) vs. comparative clinical PSD subtypes and its associated clinical conditions.

Stroke Types	Rodent Models *	Experimental Cognitive Outcome Measure	Clinical PSD Subtypes *	Associated Clinical Conditions	References **
Ischemic stroke	MCAO model	MWM, YMT, NOR, NSR, CCT [14]	Large hemispheric infarct dementia Multiple territorial infarct dementia	Intra- or extracranial atherosclerosis, or embolic source	[14,15,16,17,18,19,20,21,22,23,24,25,26,27]
	MCAO model with comorbid conditions	MWM, YMT, NOR, NSR	PSD with comorbid conditions	Aging, hypertension, diabetes, or dyslipidemia	[11,17,18,21,23,28,29,30]
	Cortical stroke model	MWM, NOR, TUNL [31], VDT [32,33,34], WWW [35], 8ARM [36], SST [37]	Small cortical infarct dementia, cortical vascular dementia, strategic infarct dementia	Single embolism, or small cortical branch occlusion	[24,25,31,32,33,34,35,36,37,38,39,40,41,42,43,44]
	Hippocampal stroke model	MWM, YMT, NOR, PAT [45], RAWMT [45], ZT [46]	Strategic infarct dementia	Cognitive pathway involvements such as thalamus, hippocampus, corpus callosum, Papez circuit, or other sophisticated areas	[14,15,16,18,19,20,24,27,31,32,35,36,42,45,46,47,48,49]
	Lacunar infarct model	NOR, SMT, PWT [49], ASST [50]	PSD with single or multiple lacunar infarcts or with underlying small vessel disease	Arteriolosclerosis, hypertensive angiopathy, single or multiple lacunar status, or vascular dementia	[49,50,51,52,53,54,55,56,57,58,59,60,61,62,63]
	Combining BCCAo and MCAO model	MWM, NOR	PSD in patients with chronic cerebral hypoperfusion	Severe carotid artery stenosis, chronic heart failure, white matter hyperintensities, subcortical ischemic vascular dementia, or impaired glymphatic clearance	[64,65,66,67]
	Stroke model with underlying AD pathology	YMT, NBT, MBT [68]	PSD in patients with underlying AD pathology	Impaired amyloid metabolism, mild cognitive impairment, AD, medial temporal lobe atrophy, brain atrophy, or mixed dementia	[59,60,61,62,63,68]
Hemorrhagic model	Intracerebral hemorrhage model	MWM, SMFT	PSD in patients with intracerebral hemorrhage	Hypertensive angiopathy, cerebral amyloid angiopathy, or cerebral microbleeds	[69,70,71,72,73,74]
	Subarachnoid hemorrhage model		PSD in patients with subarachnoid hemorrhage	Aneurysmal rupture, arteriovenous malformation, or superficial cortical hemosiderosis	[75,76,77]

Post-stroke dementia, PSD; middle cerebral artery occlusion, MCAO; bilateral common carotid artery occlusion, BCCAo; Alzheimer’s disease, AD; *, most representative rodent model among the clinical PSD subtypes; **, some referred rodent models correspond to the multiple clinical PSD subtypes; Morris water maze task, MWM; Y-maze test, YMT; novel object recognition task, NOR; novel social recognition task, NSR; contextual conditioning test, CCT; trial unique delayed non-matching to location task, TUNL; visual discrimination task-touchscreen platform, VDT; what-where-which test, WWW; 8-arm radial maze, 8ARM; set-shifting task, SST; passive avoidance test, PAT; radial arm water maze test, RAWMT; ziggurat task, ZT; paw withdrawal threshold, PWT; attention set shift test, ASST; nest-building task, NBT; marble burying task, MBT; sensorimotor function test, SMFT.

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
