# Peer review of "Rodent Models of Post-Stroke Dementia"

_ijms, 2022, doi:10.3390/ijms231810750_

Round 1

Reviewer 1 Report

In this article, Kim et al. review rodent models of post-stroke dementia. The paper is quite exhaustive, well structured, and might be able to give the potential readers an updated view of the state-of-the-art on this particular issue. Nevertheless, there are a few issues that should be addressed.

Main issues

1 Although I am aware that making a crystal-clear distinction or categorization between 1/ development, 2/ aggravation and 3/ newly manifestation of PSD is not an easy task, an additional written effort should be done in this regard. For instance, it might help having a word on how often in the clinical literature, pre-stroke clinical data of cognitive impairment/dementia are available. Otherwise, this might be a serious confounding issue for a significant part of the broad audience of Int J Mol Sci.

2 The authors exclude the BCCAO model from the review because they claim it does not show explicit/evident acute ischemic injury/stroke lesions. Despite this being partly true in a number of published studies, especially those not showing images of the actual lesioned brain tissue, others do report lesions and show such images using different dyeing techniques, e.g. TTC for gross assessment in brain slices, or others for microscopy (Nissl, H&E, immunofluorescence). Moreover, most of these articles show convincing data of cognitive impairment using several biobehavioural tests.

3 It would have been appreciated to read the authors’ conclusion on whether the MCAO and the lacunar stroke models might be considered good PSD models in rodents.

Minor issues

1 The authors might consider changing the expression ‘animal models’ to ‘rodent models’ throughout the text (abstract, keywords and the main text), for the authors’ interest is centered in this group of animals, as they state in the title itself.

2 Abbreviate ‘MCAO’ the first time it appears in full in the abstract.

3 Is it necessary to list in the abstract all the rodent PSD models that will be reviewed thereafter in the main text?

4 The word ‘amyloid’ as a keyword would be more precise if it was accompained by plaque, beta, deposition, aggregation etc.

5 Since a great deal of the review focuses on rodent models of ischemic stroke, I would suggest including a keyword in this regard.

6 The sentence ‘Any dementia manifested…’ (lines 31-33) is rather forced; I suggest to rewrite it.

7 Line 34: ‘seem’; ‘the clinical diagnosis and classification…’, of PSD?

8 Should the authors considered it convenient, the Introduction might benefit from the reading, and the eventual inclusion in the bibliography, of a couple of papers: Weaver et al, published in Lancet Neurol a year ago or so, and Kim et al, Int J Mol Sci, 23(2):602, 2022.

9 Line 90: ‘novel object test’.

10 Line 197: ‘popular’ is not the most adequate adjective here in this context.

11 Line 232: write ‘AQP4’ in full.

Author Response

Please, find attached file.

Reviewer 2 Report

The review of Kim at el. deals with the topic of experimental modelling of post-stroke dementia. It seems to me, however, that the paper on this subject should provide a broader view on PSD and related issues.

First of all, the review of the available literature concerning the models is rather limited and does not exhaust the topic. The paper is focusing only on listing the models and providing a short description. The current structure of the manuscript is unfortunately rather confusing, there are sections that refer to the technique (MCAO, BCCAO and MCAO), to the structure (hippocampal stroke model) or a certain setting (stroke model with comorbid conditions), this should be changed and unified.

Importantly, this review includes only little insight into proposed mechanisms in the experimental models, a paragraph with more details would be of added value (e.g. what about the paper of Doyle et al. 2015 postulating an immune-mediated mechanism?). Also, from the experimental perspective, adding a more extensive overview of tests used to detect PSD in experimental setting would be great.

Additional points/minor issues:

In the abstract, the MCAO model is mentioned twice.

Not only the hippocampus, but other specific brain areas also playing crucial role in cognition can be critical to the development of PSD. Involvement of cognitive pathways, such as thalamus, hippocampus, corpus callosum, Papez circuit, or other sophisticated areas can result into the development of PSD, even if it is a very regional stroke.” – this fragment should be rewritten, it is a bit vague.

Underlying pathological brain conditions, such as chronic cerebral hypoperfusion or AD pathology, can be vulnerable to the development of PSD.” – this sentence requires some revision, “vulnerable” seems not to be the best word here.

Line 210 – should be “novel object (recognition) test”?

Line 253 “These experimental results suggest that the presence of amyloid pathology may be a strong risk factor for developing PSD, which is consistent with the clinical finding that stroke patients with underlying amyloid pathology could develop a more severe and rapid cognitive decline over three years than those without underlying amyloid pathology” – this fragment contains some redundancies and should be revised.

Line 328  “In our study, this phenomenon was significantly enhanced when cerebral hypoperfusion was combined” – combined with what? Please revise.

Author Response

Please, find the attached file.

Round 2

Reviewer 1 Report

The issues I found in the MS have been adequately addressed.

Author Response

Thanks for your review.